

# A systematic review and meta-analysis of magnetic resonance and computed tomography enterography in the diagnosis of small intestinal tumors

Ruitao Li[*], Shengqiang Ye[*], Chenglong Zhou, Feng Liu and Xiaonan Li

Department of Radiology, Shengli Oilfield Central Hospital, Dongying, China
[*] These authors contributed equally to this work.

## ABSTRACT

**Objective**. To explore the potential value of magnetic resonance (MR) and computed tomography (CT) enterography in the diagnosis of small intestinal tumor (SIT).

**Methods**. Articles reporting on the diagnosis of SIT by MR and CT enterography deposited in Chinese and foreign literature databases were identified and evaluated using the quality assessment of diagnostic accuracy studies (QUADAS). The diagnostic data extracted from the articles were adopted for meta-analysis using Meta-disc 1.40 software. Analysis was undertaken to compare the sensitivity, specificity, positive and negative likelihood ratios, and the diagnostic odds ratio (DOR) of MR and CT enterography in the diagnosis of SIT. The diagnostic values of the two imaging methods were analyzed by summary receiver operating characteristic (SROC) curves. The meta-analysis was registered at INPLASY (202380053).

**Results**. A total of eight articles, including 551 cases of SIT were included in this analysis. The pooled sensitivity and specificity of MR enterography were 0.92 (95% CI [0.89–0.95]) and 0.81 (95% CI [0.74–0.86]), respectively, whilst CT enterography had a sensitivity of 0.93 (95% CI [0.90–0.95]) and a specificity of 0.83 (95% CI [0.76–0.88]). For MR enterography, the combined positive likelihood ratio was 4.90 (95% CI [3.50–6.70]), the combined negative likelihood ratio was 0.10 (95% CI [0.07–0.14]), and the area under the receiver operating characteristic curve (AUROC) was 0.940. For CT enterography, the corresponding values were 5.40 (95% CI [3.90–7.40]), 0.08 (95% CI [0.06–0.12]), and 0.950, respectively. When the pretest probability for MR was assumed to be 50%, the posterior probabilities for positive and negative results were calculated as 83% and 9%, respectively. For CT enterography with a pretest probability of 50%, the posterior probabilities of positive and negative results were 84% and 8%, respectively.

**Conclusion**. MR and CT enterography have high accuracy in the diagnosis of SIT and have a valuable role in the diagnosis and management of these tumors.

Corresponding author
Ruitao Li, lrtjt0326@126.com

## INTRODUCTION

The length of the small intestine and mucosal surface area account for 75–90% of the alimentary tract yet primary tumors in the small intestine account for only 1–5% of gastrointestinal cancers (*Bhalla et al., 2013*). The diagnosis and treatment of small intestinal tumors (SITs) is often delayed due to the lack of early detection biomarkers and the late presentation of clinical symptoms. Lesions are often detected at the advanced stage of the disease when abdominal masses can be palpated or when the intestine is obstructed (*Assumpção et al., 2020a*).

The small intestine is a coiled tissue that overlaps in the abdominal cavity and is subject to high levels of movement that is challenging during imaging examinations. The gastrointestinal barium meal is the first choice for imaging examinations and computed tomography (CT) enterography plays an increasingly prominent role in the diagnosis and evaluation of intestinal diseases. Improvements in CT technology have allowed improved detection of lesions in the intestinal wall and cavity (*Guglielmo et al., 2020*; *Grajo et al., 2021*). Magnetic resonance imaging (MRI) does not rely on ionizing radiation for imaging and so is appropriate for pregnant women and children who are unsuitable for CT imaging. Compared to CT, MRI also has high soft tissue resolution and can display the anatomical details of internal, parietal and external small intestines. For example, T1-weighted SIT or lipomas containing fat components show high signals and T2-weighted hemangiomas show obvious high signals (*Bruining et al., 2018*; *Horvat et al., 2018*).

Previous studies have reported the application of MR and CT enterography in the diagnosis of SIT (*Yang, 2019*; *Fu et al., 2016*; *Zhao, 2017*), yet these studies draw inconsistent conclusions. In this study, we objectively evaluated the role of MR and CT enterography in the diagnosis of SIT through the comprehensive retrieval and screening of published studies to provide a basis for the selection and formulation of clinical treatment options in SIT.

## MATERIALS & METHODS

### Inclusion and exclusion criteria

The inclusion criteria for articles included in the analysis were (1) controlled experiments or the diagnostic study of MR and/or CT enterography in the diagnosis of SIT; (2) data reporting true positive (TP), false positive (FP), true negative (TN) and false negative (FN) cases extracted directly or indirectly from the table (in the form of 2 × 2); (3) studies with original data to estimate the kappa value and its standard error; (4) histopathological examinations used as the gold standard for patient diagnosis; (4) studies including >10 cases; and (5) studies published in Chinese or English language.

The exclusion criteria were (1) duplicate reports and articles that did not offer original data and evidence of interest; and (2) case reports, letters, comments, cell and animal experiments, reviews, and non-relevant studies.

### Retrieval strategy for articles

The Cochrane Library, PubMed, EMBASE, CINAHL, VIP, Wanfang and CNKI databases were searched using a computer to collect articles on the diagnosis of SIT by MR and

(or) CT enterography from the establishment of the database to August 14th, 2023. The retrieval words in English were MR, magnetic resonance spectroscopy, magnetic resonance spectroscopies, MR spectroscopy, magnetic resonance, NMR spectroscopy, NMR spectroscopies, spectroscopies, NMR, computed tomography angiography, angiographies, computed tomography, computed tomography angiographies, angiography, CT, CT angiography, CT angiographies and small intestine cancer. The retrieval words in Chinese were magnetic resonance, CT enterography, small intestinal tumor, small intestinal carcinoma, primary small intestine carcinoma, malignant tumor of the small intestine and primary small intestine carcinoma. To minimize the omission of articles, this study combined manual retrieval using a combination of keywords and subject words with language limited to Chinese and English.

## Literature screening and data extraction

Literature screening and data extraction were conducted independently by two researchers (Shengqiang Y and Chenglong Z) and consensus was reached by joint discussion when differences arose. Irrelevant studies were excluded by browsing the titles and abstracts followed by reading the full text to determine if the study should be included. Relevant information was extracted from each article including the name of the first author, publication date and sample size. The meta-analysis was registered at INPLASY (International Platform of Registered Systematic Review and Meta-analysis Protocols, 202380053). The study was approved by the Institutional Review Board and Research Ethics Committee of the Shengli Oilfield Central Hospital.

## Literature quality evaluation

The researchers independently completed a literature quality evaluation. For studies with inconsistent results, the quality assessment of diagnostic accuracy studies (QUADAS) (*Yang et al., 2021*) was used to evaluate the quality of the included studies after discussions. Each criterion was divided into three levels, namely, "yes", "no" and "unclear". Amongst them, "yes" referred to meeting this criterion, "no" was unsatisfied, and "unclear" was unable to obtain sufficient information from the text.

## Statistical analysis

Meta-Disc 1.4 and Stata software were used for statistical analysis. The publication bias of articles was detected by the Egger method and the $\chi^2$ test was used to analyze the heterogeneity of the diagnostic ratio in each study. The kappa value was estimated from the positive and negative test results. The standard error (SE) and a 95% confidence interval (CI) for kappa were then calculated. Using the fixed effect model, it was found that I2 < 50% and $P > 0.05$, indicating no heterogeneity. Conversely, the random effect model detected heterogeneity for I2 $\geq$ 50% and $P < 0.05$. Examination of heterogeneity sources, including threshold and non-threshold effects, was conducted. Additionally, a meta-analysis was performed on all included articles to compute the combined sensitivity, specificity, and AUROC curve. All findings were presented with a 95% confidence interval, and statistical significance was established for $P$-values < 0.05.

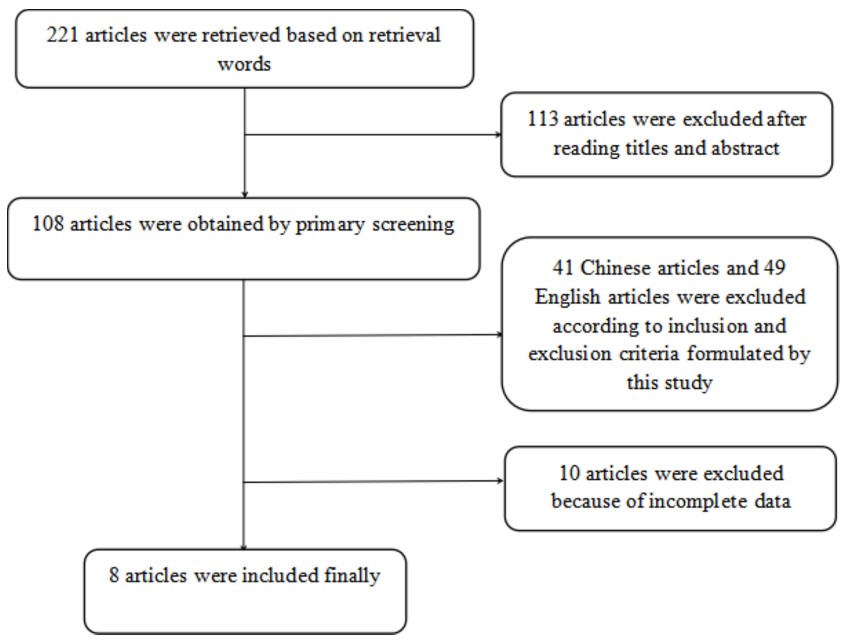

**Figure 1** Screening process of included articles.

## RESULTS

### Retrieval results of articles

A total of 221 articles were initially retrieved based on the selected keywords and after excluding studies that did not meet the inclusion criteria (see Fig. 1), eight articles (*Yang, 2019*; *Fu et al., 2016*; *Zhao, 2017*; *Zhuang et al., 2018*; *Chen, Ren & Huang, 2017*; *Wang et al., 2015*; *Zhang, 2018*; *Feng et al., 2022*) were deemed eligible for inclusion in the meta-analysis. The characteristics of the studies included in the analysis are summarized in Table 1. Of these articles, six examined the concurrent utilization of MR and CT enterography in the diagnosis of SITs, whilst two studies compared the diagnostic efficacy of MR and CT enterography individually and in combination.

### Quality control

The quality evaluation tool of QUADAS was used to evaluate literature quality as shown in Table 2. The items related to bias included the bias of disease progression (item 4), bias of multiple references (item 6), mixed bias (item 7) and bias of experimental interpretation (item 10), whose coincidence rates of "yes" were 100%, and bias of partial references (item 5) was "yes" in 80% of articles, showing a small possibility of bias. The coincidence rates of Items 11 and 12 for "no" and "unclear" were 80% and 60%, respectively. These data indicated that the gold standard was quite different from clinical practice in diagnostic interpretation of the results.

In the coincidence rates of items 1, 2 and 3, "yes" was 100%, indicating the strict criteria for screening patients and diagnosing the disease spectrum to clearly propose the inclusion and exclusion criteria of patients by the same reference standard. The coincidence rates of

**Table 1  Basic characteristic of included articles.**

| Articles | Publishing time (years) | Age (years old) | Sample sizes | MR | | | | CT enterography | | | | Gold standard |
|---|---|---|---|---|---|---|---|---|---|---|---|---|
| | | | | TP | FP | TN | FN | TP | FP | TN | FN | |
| *Feng et al. (2022)* | 2022 | 21–58 | 70 | 48 | 3 | 4 | 15 | 49 | 3 | 5 | 17 | Pathology |
| *Wang et al. (2015)* | 2015 | 15–72 | 65 | 46 | 4 | 3 | 12 | 46 | 4 | 2 | 13 | Pathology |
| *Chen, Ren & Huang (2017)* | 2017 | 34–68 | 58 | 34 | 4 | 5 | 15 | 33 | 4 | 4 | 17 | Pathology |
| *Zhuang et al. (2018)* | 2018 | 18–68 | 60 | 30 | 5 | 3 | 22 | 30 | 4 | 3 | 23 | Pathology |
| *Zhang (2018)* | 2018 | 34–69 | 50 | 32 | 3 | 2 | 13 | 32 | 2 | 2 | 14 | Pathology |
| *Zhao (2017)* | 2017 | 22–79 | 96 | 71 | 4 | 5 | 16 | 72 | 4 | 3 | 17 | Pathology |
| *Yang (2019)* | 2019 | 45–71 | 74 | 41 | 3 | 5 | 25 | 39 | 4 | 5 | 26 | Pathology |
| *Fu et al. (2016)* | 2016 | 46–69 | 78 | 55 | 5 | 3 | 15 | 47 | 5 | 2 | 16 | Pathology |

**Notes.**
CT, computed tomography; MR, magnetic resonance; TP, true positive; FP, false positive; TN, true negative; FN, false negative.

implementing quasi-evaluation test (item 8) and the gold standard (item 9) were 100% indicating that the report on quasi-evaluation test and the gold standard test was well described. However, in the implementation process, the treatment of intermediate results or existing studies was poor causing the coincidence rates of items 13 and 14 for "yes" as 60% and 20%, respectively, showing a greater possibility of bias.

## Results of the meta-analysis

### Accuracy of MR and CT enterography in the diagnosis of SIT

All cases with $P$-values > 0.05 in the heterogeneity test (diagnostic sensitivity and specificity of MR as 0.92 and 0.81, and diagnostic sensitivity and specificity of CT enterography as 0.93 and 0.83) indicated no heterogeneity amongst the articles and so a fixed effect model was used for meta-analysis as shown in Fig. 2.

### Results of image analysis between the two groups

Table 3 shows the combined sensitivity, specificity, positive likelihood ratio, negative likelihood ratio and diagnostic odds ratio, and the AUROC of MR and CT enterography in the diagnosis of SIT.

### SROC curve of MR and CT enterography in the diagnosis of SIT

The AUCs were 0.940 for MR and 0.950 for CT enterography indicating that the accuracy rate of CT enterography was slightly higher than for MR in the diagnosis of SIT, as detailed in Fig. 3.

### Analysis of publication bias

The symmetrical Deek's funnel plot showed no publication bias. The asymmetry test results of MR and CT enterography in the diagnosis of SIT showed that the $P$ values were 0.54 and 0.55, respectively, as shown in Fig. 4.

### Posterior probability

A Fagan plot was drawn using Stata 17.0 software. When the pretest probability was 25%, the correct rate of positive MR for the diagnosis of SIT was 62%, whilst only 3% of negative

**Table 2  Quality evaluation tool of QUADAS to evaluate literature quality.**

| Evaluation criteria | Yes | No | Unclear |
|---|---|---|---|
| 1. Whether the case spectrum included various cases and easily confused disease cases | 5 | 0 | 0 |
| 2. Whether the selection criteria of research objects were clear | 5 | 0 | 0 |
| 3. Whether gold standard could distinguish the states of disease and health | 5 | 0 | 0 |
| 4. Whether the interval between the gold standard and quasi-evaluation standard was short enough to avoid a change in disease condition | 5 | 0 | 0 |
| 5. Whether all samples or randomly selected samples received gold standard test | 4 | 1 | 0 |
| 6. Whether all cases were subjected to gold standard test, regardless of the outcomes of quasi-evaluation test | 5 | 0 | 0 |
| 7. Whether gold standard test was independent of quasi-evaluation test | 5 | 0 | 0 |
| 8. Whether the operations of quasi-evaluation test were described clearly enough and could be repeatable | 5 | 0 | 0 |
| 9. Whether the operations of gold standard test were described clearly enough and could be repeatable | 5 | 0 | 0 |
| 10. Whether the results interpretation of quasi-evaluation test was carried out without knowledge of gold standard test results | 5 | 0 | 0 |
| 11. Whether the results interpretation of gold standard test was carried out without knowledge of results of quasi-evaluation test | 1 | 4 | 0 |
| 12. Whether available clinical data were consistent with the clinical data in practice when interpreting trial results | 2 | 0 | 3 |
| 13. Whether intermediate test results with difficulties in interpretation were reported | 3 | 2 | 0 |
| 14. Whether the literature account for cases that withdrew from the experiment | 1 | 3 | 1 |

patients may be diagnosed with SIT. When the pretest probabilities were 50% and 75%, the posterior probabilities of positive MR were 83% and 94%, and the posterior probabilities of negative MR were 9% and 22%, respectively. When the pretest probability was set to 25%, the correct rate of positive CT enterography for the diagnosis of SIT was 64%, whilst only 3% of negative patients were diagnosed with SIT. The posterior probabilities of positive MR were 84% and 94% when pretest probabilities were 50% and 75%, and the negative posterior probabilities were 8% and 20%, as shown in Fig. 5.

## DISCUSSION

An epidemiological survey showed that the incidence of SIT is 1.1/10 million, accounting for only 1/10 of the incidence of colon cancer (*Assumpção et al., 2020b*). The low incidence of SIT compared to other parts of the gastrointestinal tract may be due to the following reasons; (1) the small intestine contains fluids that are weakly oncogenic and dilute potential

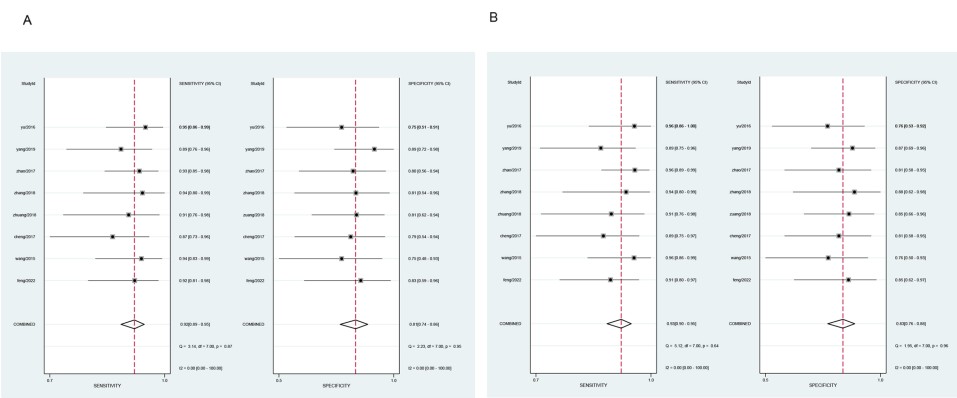

**Figure 2  Meta-analysis of the sensitivity and specificity of MR and CT enterography in the diagnosis of SIT.** (A) and (B) showed meta-analysis of sensitivity and specificity of MR and CT enterography in the diagnosis of SIT. CT, computed tomography; MR, magnetic resonance; SIT, small intestinal tumor.

**Table 3  Indexes comparison of MR and CT enterography in the diagnosis of SIT.**

| Diagnostic methods | Combined sensitivity (95% CI) | Combined Specificity (95% CI) | Combined positive likelihood ratio (95% CI) | Combined negative likelihood ratio (95% CI) | Combined diagnostic odds ratio (95% CI) | AUROC |
|---|---|---|---|---|---|---|
| MR | 0.92 (0.89∼0.95) | 0.81 (0.74∼0.86) | 4.90 (3.50∼6.70) | 0.10 (0.07∼0.14) | 51 (30∼88) | 0.940 |
| CT enterography | 0.93 (0.90∼0.95) | 0.83 (0.76∼0.88) | 5.40 (3.90∼7.40) | 0.08 (0.06∼0.12) | 64 (36∼112) | 0.950 |

**Notes.**

CT, computed tomography; MR, magnetic resonance; SIT, small intestinal tumor; AUROC, area under the receiver operating characteristic curve.

A

B

**Figure 3  SROC curve of MR and CT enterography in the diagnosis of SIT.** (A) and (B) showed MR and CT enterography, respectively. CT, computed tomography; MR, magnetic resonance; SIT, small intestinal tumor.

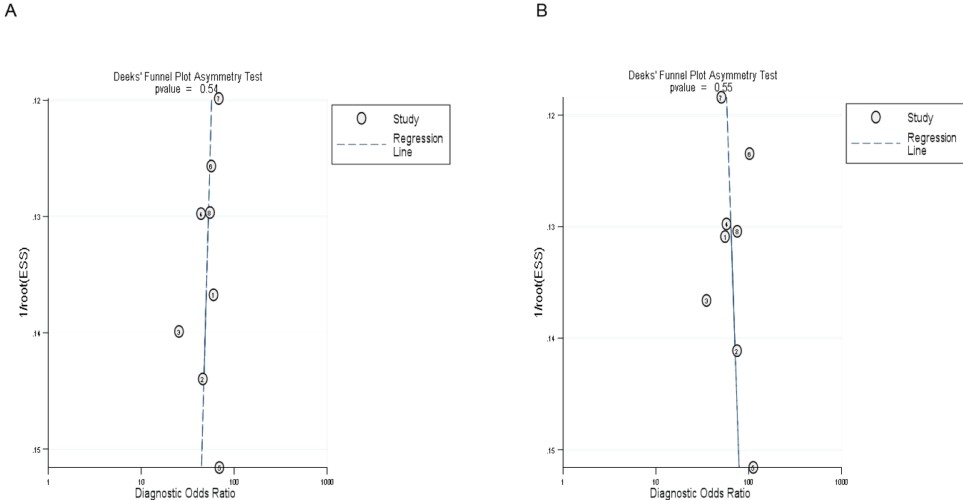

**Figure 4   Deek's funnel plot of MR and CT enterography in the diagnosis of SIT.** (A) and (B) showed MR and CT enterography, respectively. CT, Computed tomography; MR, Magnetic resonance; SIT, Small intestinal tumor.

carcinogens. Also, the rapid peristalsis of the small intestine greatly reduces the contact time between carcinogens and the mucosa; (2) the fluid in the small intestinal is alkaline and has a high concentration of benzopyrene hydroxylase that could potentially inactivate certain carcinogens to prevent tumor formation; (3) lower flora in the small intestine reduces the participation of anaerobic bacteria in the metabolism of cholic acid to reduce the levels of potential carcinogens; (4) the lymphatic tissues of the small intestine are the main sites for the production of IgA, and high concentrations of IgA can neutralize viruses and other carcinogens; and (5) the lymph nodes of the small intestine contain mainly T lymphocytes that have strong immunity and anti-tumor growth characteristics.

Previous studies have shown that SIT occur in various tissues of the small intestine (*Baiu & Visser, 2019*; *Sanchez-Mete & Stigliano, 2019*). The most common benign tumor is an adenoma, followed by leiomyoma, whilst adenocarcinoma is the most common malignant tumor, followed by carcinoid tumors, leiomyosarcoma and lymphoma. At present, the diagnosis of SIT includes localization and qualitative diagnosis. The main reasons for the low rate of diagnosis before surgery include low morbidity, limited vigilance and limited specific symptoms that are confused with other digestive tract diseases. Overall, there is currently a lack of early detection biomarkers for diseases of the small intestine (*Alfagih, Alrehaili & Asmis, 2022*).

The current examination methods for SIT include enterography, radionuclide imaging, abdominal CT, MR, capsule endoscopy and double balloon enteroscopy (*Raman & Fishman, 2016*). MR provides a high resolution of soft tissues and can provide clear images of lesions in and outside of the lumen and the intestinal wall. However, due to large scanning ranges, the intestinal lumen requires a contrast agent that can result in imaging artefacts due to respiratory movements and image pleats (*Fidler et al., 2017*). In contrast, CT enterography displays the morphology of lesions in the small intestine and

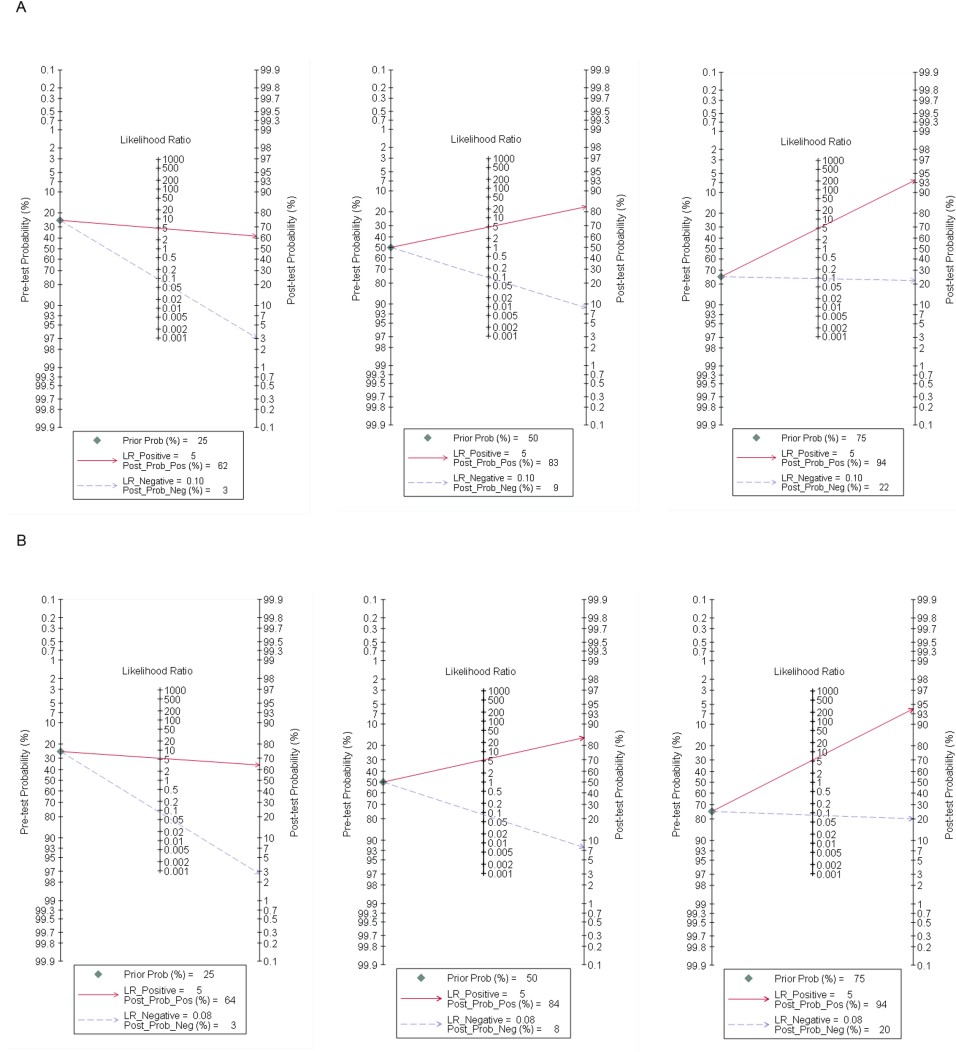

**Figure 5   Fagan plot of MR and CT enterography in the diagnosis of SIT.** (A) and (B) showed MR and CT enterography, respectively. CT, computed tomography; MR, magnetic resonance; SIT, small intestinal tumor.

can display structural relationships between lesions and surrounding tissues *via* multi-layer and thin-layer scanning and reconstruction (*Horton & Fishman, 2003*). The injection of contrast agents to fill the intestinal cavity enhances and displays the contour of diseased tissues and improves the resolution of CT images of soft tissues. Multi-phase scanning is helpful in displaying the details of small intestinal lesions (*Liu et al., 2006*) which can help to carefully identify small lesions.

A previous report compared the studies on MR and CT enterography in the diagnosis of SIT (*Li et al., 2018*) yet the conclusions were inconsistent. In the current study, data were collected from relevant articles on the diagnosis of SIT in Chinese and foreign literature databases and meta-analysis was used to quantitatively summarize the imaging methods in the diagnosis of SIT. The values of MR and CT enterography in the diagnosis of SIT were

compared to comprehensively evaluate the clinical value of the imaging methods. When the AUROC curve was ≤0.5, there was no diagnostic value. In this study, the area under the SROC curves for MR and CT enterography were 0.940 and 0.950, respectively, which was similar to the results of other studies (*Van Weyenberg et al., 2010*). These data suggest that both MR and CT have high diagnostic accuracy for SIT.

The presented meta-analysis included eight articles on MR and CT enterography in the diagnosis of SIT. The results showed that the combined sensitivity and specificity of MR in the diagnosis of SIT were 0.92 (95% CI [0.89–0.95]) and (95% CI [0.74–0.86]), whilst the comparative values for CT enterography were 0.93 (95% CI [0.90–0.95]) and 0.83 (95% CI [0.76–0.88]), indicating a higher diagnostic efficiency. The combined negative likelihood ratios were 0.10 (95% CI [0.07–0.14]) and 0.08 (95% CI [0.06–0.12]) suggesting that malignancy could be excluded when the diagnosis was negative. A Fagan plot showed that when the pretest probability was 50% in MR, the posterior probabilities of positive and negative MR were 83% and 9%, and 84% and 8% for CT enterography. Assuming that the probability of clinicians to diagnose SIT according to clinical manifestations and personal experience is 50%, these data indicate that the accuracy of SIT diagnosis increased from 50% to 83% with positive findings after MR examination. If the MR examination results were negative, the possibility of patients suffering from SIT decreased from 50% to 9%. When CT enterography was performed, the accuracy of SIT diagnosis increased from 50% to 84% if the result was positive. If the result of CT enterography was negative, the possibility of SIT diagnosis decreased from 50% to 8%. These data show that MR and CT have high clinical accuracy in the diagnosis of SIT.

The findings presented in this study have significant implications for the application of MR and CT enterography in the diagnosis of SIT. Firstly, the study found that MR and CT enterography have high diagnostic accuracy for SIT as indicated by the areas under the SROC curves. The AUCs for MR and CT enterography were 0.940 and 0.950, respectively. These data suggest that both imaging modalities can effectively differentiate between benign and malignant small intestinal tumors.

Secondly, the study demonstrated that MR and CT enterography have comparable diagnostic efficacy in SIT. The combined sensitivity and specificity of MR in the diagnosis of SIT were 0.92 and 0.81, respectively, while for CT enterography, the values were 0.93 and 0.83. The combined negative likelihood ratios for MR and CT enterography were 0.10 and 0.08, respectively. These findings indicate that both imaging techniques can provide valuable information for ruling out SIT when the diagnosis is negative.

Moreover, this study showed that MR and CT enterography can significantly improve the accuracy of diagnosis compared to clinical judgment alone. The Fagan plot analysis demonstrated that when the pretest probability of SIT was 50%, a positive result from MR examination increased the posterior probability of SIT to 83%, whilst a negative result decreased the probability to 9%. Similarly, for CT enterography, a positive result increased the posterior probability of SIT to 84%, whilst a negative result decreased it to 8%. These results indicate that MR and CT enterography can help clinicians include or exclude a diagnosis of SIT with a higher level of accuracy.

The application of MR and CT enterography in SIT is grounded in their ability to provide detailed and comprehensive imaging of the small intestine (*Cai et al., 2023*). Given the limitations of traditional diagnostic methods, such as low morbidity, limited vigilance from doctors, and a lack of simple and non-invasive techniques, MR and CT enterography offer valuable clinical potential (*Hu et al., 2022*). Both MR and CT enterography offer high-resolution imaging, enabling the visualization of small intestinal lesions from multiple perspectives and planes (*Li et al., 2023*). MR enterography excels in providing clear images of the small intestinal lumen, external structures, and intestinal walls, whilst CT enterography allows the accurate display of morphological details and anatomical relationships between lesions in the small intestine and surrounding tissues (*Ma et al., 2022*; *Wang, Chen & Hu, 2022*).

Advancements in imaging technologies and analysis including artificial intelligence (AI) improved the accuracy and efficiency of diagnostic methods. AI algorithms can be trained to assist radiologists in analyzing medical images from MR and CT scans. These algorithms can help to identify and characterize abnormalities such as SIT by analyzing the image data. AI can aid in detecting subtle features, measuring tumor size, assessing the extent of the disease, and providing additional insights to assist in diagnosis. The integration of AI into MR and CT enterography has the potential to enhance the detection and diagnosis of SIT by providing a more objective and consistent analysis of the imaging findings.

The data presented in this study are subject to several limitations. Firstly, the language inclusion criteria were limited to Chinese and English resulting in potential biases. Also, the inability to access unpublished literature may further contribute to these biases. Secondly, the varying quality of the included articles also influenced the analysis. The observed heterogeneity may be attributed to factors such as differences in research study designs and disease progression. Despite these limitations, these findings align with previous research and provide evidence of the diagnostic value and comparable efficacy of MR and CT enterography in evaluating SIT.

## CONCLUSIONS

The diagnosis of SIT using MR and CT can provide important clinical information and inform treatment decisions. MR and CT have equivalent diagnostic efficacy for SIT and these approaches can reduce the need for puncture biopsy in patients and provide a more reliable basis for the diagnosis and treatment of SIT.

### Funding
The authors received no funding for this work.

### Competing Interests
The authors declare there are no competing interests.

## Author Contributions

- Ruitao Li conceived and designed the experiments, analyzed the data, authored or reviewed drafts of the article, and approved the final draft.
- Shengqiang Ye conceived and designed the experiments, analyzed the data, authored or reviewed drafts of the article, and approved the final draft.
- Chenglong Zhou performed the experiments, prepared figures and/or tables, and approved the final draft.
- Feng Liu performed the experiments, prepared figures and/or tables, and approved the final draft.
- Xiaonan Li performed the experiments, analyzed the data, prepared figures and/or tables, and approved the final draft.

## Human Ethics

The following information was supplied relating to ethical approvals (i.e., approving body and any reference numbers):

The study was approved by the Institutional Review Board and Research Ethics Committee of the Shengli Oilfield Central Hospital.

## Data Availability

The raw data is available in the Supplementary File.

## Supplemental Information

Supplemental information for this article can be found online at http://dx.doi.org/10.7717/peerj.16687#supplemental-information.

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
