# Peer review of "A systematic review and meta-analysis of magnetic resonance and computed tomography enterography in the diagnosis of small intestinal tumors"

_PeerJ, doi:10.7717/peerj.16687_

## Round 0.1 · original submission · Major Revisions

Please carefully revise according to the Reviewers' comments.

**Language Note:** The review process has identified that the English language must be improved. PeerJ can provide language editing services - please contact us at copyediting@peerj.com for pricing (be sure to provide your manuscript number and title). Alternatively, you should make your own arrangements to improve the language quality and provide details in your response letter. – PeerJ Staff

Reviewer 1 ·

Basic reporting

The manuscript is interesting and well-written. The authors fully introduced the background of the article and provided a detailed explanation of the conclusion of the paper

Experimental design

The manuscript is interesting and well-written. The application of MR and CT enterography in the diagnosis of small intestinal tumor were investigated. The results are solid and indicate that patients in each subgroup should receive distinct personalized treatment. The manuscript can be considered for publication after addressing the following questions:
1. Full name for the abbreviation should be exhibited at the first presence.
2. The result “Accuracy of MR and CT enterography in the diagnosis of SIT” in abstract conclusion differs from results, please correct.
3. The Discussion could be enhanced by addressing the implications of the study's findings in the context of current literature on application of MR and CT enterography in small intestinal tumor.
4. Discussing potential limitations and biases, as well as avenues for future research, would contribute to a more well-rounded conclusion.
5. It will be better to show kappa for the selection and data extraction. Please show the data of kappa of agreement during the systematic searches.
6. Is there any innovation in this article compared to the previous article?
7. It is not clear on which basis studies/reports were included and why others (a lot of studies, in fact) have been excluded.
8. The discussion of AI could be more specific about how MR and CT enterography helps with detecting small intestinal tumor.

Validity of the findings

no comment

Additional comments

no comment

Reviewer 2 ·

Basic reporting

1. Please note that the reviewers have deemed the quality of language in this manuscript unsuitable for publication. Please send the manuscript to a language editing company to improve the article for language and style. Please provide the certificate confirming that language editing has been performed at the same time as the response to the peer review comments.
2. It would be beneficial to include a brief rationale for the application of MR and CT enterography in small intestinal tumor and the limitations of current diagnostic methods. This would provide readers with a clearer understanding of the study's significance.
3. The method of this study is not novel enough. Authors need to emphasize their innovative contributions.
4. Several sentences are quite long and complex, leading to a lack of clarity and coherence.
5. Please attach the cited literature for the method section.
6. The model of meta-analysis should be confirmed, such as fixed or random.
7. Please add the full name of the abbreviations in tables and figures.
8. In the abstract, the 95% CI of each pooled indicator should be provided.

Experimental design

See above

Validity of the findings

See above

Additional comments

See above

---

## Round 0.2 · accepted · Accept

Well done. It could be accepted in this journal.

Reviewer 1 ·

Basic reporting

Well done

Experimental design

Well done

Validity of the findings

Well done

Additional comments

no

Reviewer 2 ·

Basic reporting

The authors have replied well and addressed concerns.

Experimental design

see above

Validity of the findings

The authors have replied well and addressed concerns.

Additional comments

None.